# Superior polarization retention through engineered domain wall pinning

Dawei Zhang[1,6], Daniel Sando [1,2,3,6]*, Pankaj Sharma [1,2]*, Xuan Cheng[4], Fan Ji[1,2], Vivasha Govinden[1,2], Matthew Weyland [4,5], Valanoor Nagarajan[1,2] & Jan Seidel[1,2]*

Ferroelectric materials possess a spontaneous polarization that is switchable by an electric field. Robust retention of switched polarization is critical for non-volatile nanoelectronic devices based on ferroelectrics, however, these materials often suffer from polarization relaxation, typically within days to a few weeks. Here we exploit designer-defect-engineered epitaxial $BiFeO_3$ films to demonstrate polarization retention with virtually no degradation in switched nanoscale domains for periods longer than 1 year. This represents a more than 2000% improvement over the best values hitherto reported. Scanning probe microscopy-based dynamic switching measurements reveal a significantly increased activation field for domain wall movement. Atomic resolution scanning transmission electron microscopy indicates that nanoscale defect pockets pervade the entire film thickness. These defects act as highly efficient domain wall pinning centres, resulting in anomalous retention. Our findings demonstrate that defects can be exploited in a positive manner to solve reliability issues in ferroelectric films used in functional devices.

[1] School of Materials Science and Engineering, UNSW Sydney, Sydney, NSW 2052, Australia. [2] ARC Centre of Excellence in Future Low-Energy Electronics Technologies, UNSW Sydney, Sydney, NSW 2052, Australia. [3] Mark Wainwright Analytical Centre, UNSW Sydney, High Street, Kensington, NSW 2052, Australia. [4] Department of Materials Science and Engineering, Monash University, Melbourne, VIC 3800, Australia. [5] Monash Centre for Electron Microscopy, Monash University, Melbourne, VIC 3800, Australia. [6] These authors contributed equally: Dawei Zhang, Daniel Sando. *email: daniel.sando@unsw.edu.au; pankaj.sharma@unsw.edu.au; jan.seidel@unsw.edu.au

Ferroelectric materials, characterized by a spontaneous polarization that can be switched by an external electric field, are currently widely investigated for developing low-voltage non-volatile nanoelectronics[1–3]. Distinct directional polarization states ($+P$ and $-P$) in ferroelectrics can represent the computational 0 and 1 states used in binary systems, utilized, e.g., in non-volatile ferroelectric random access memory (FeRAM). Ongoing research on FeRAM[4–8] technology and related nanoscale ferroelectric devices[9–14] has recently attracted significant interest. Investigations based on scanning probe microscopy (SPM) have shown that the information bit writing and data reading can be scaled down to the nanometre scale, enabling miniaturization[6,13,15]. Polarization retention, which dictates the lifetime of stored information in ferroelectric materials, i.e., the stability of the originally written polarization direction and magnitude over time, is a crucial performance feature of such technologies. Retention loss, the time-dependent decay of the polarization, in this context renders ferroelectricity unstable and can lead to associated information storage failure. The retention problem is exacerbated in SPM probe-based tip-ferroelectric-electrode configurations because of the typically asymmetric electrostatic boundary conditions originating from built-in fields, residual depolarization fields, and internal/external space charges[16–18].

Considerable research[5,6] has been dedicated to the retention problem in many standard ferroelectric systems such as $SrBi_2Ta_2O_9$[16,19], $Pb(Zr,Ti)O_3$[17,18,20–23], $PbTiO_3$[24], $LiNbO_3$[25], $BaTiO_3$[26,27], and $BiFeO_3$ (BFO)[28,29]. The origin of the retention problem is presumed to be associated with depolarization fields (as a result of incomplete compensation of polarization bound charges) and built-in fields (due to work function difference near the film/electrode interface)[16,17,22]. Very recently, BFO has attracted significant interest as a promising ferroelectric material to improve the retention loss issue. Researchers have found that in this system a giant enhancement of polarization retention of more than 450 h can be realized at mixed-phase boundaries (termed $R/T$ phases in ref. [30], whereas in this study we refer to it as $R'/T'$ phases, where $T'$ denotes a tetragonal-like phase and $R'$ denotes a rhombohedral-like phase)[30]. It is suggested that the in-plane (IP) periodic elastic potentials at the $R/T$ mixed boundaries act as pining centres to maintain domain stability. In another work, (111)-oriented BFO mesocrystals were grown within a stiff matrix of $CoFe_2O_4$[31]. This matrix can mechanically clamp the BFO mesocrystals, thereby suppressing the ferroelectric relaxation process. In this study, a defect engineering method is used to design and fabricate a special BFO thin film that is not susceptible to retention loss over time, shedding light on resolving this long-standing problem.

Our work is based on epitaxial BFO thin films grown on $LaAlO_3$ (LAO) substrates, where the ferroelectric domain structures are largely determined by the strain state originating from the lattice mismatch between the thin film and the substrate. In general, increasing the film thickness to ~30 nm triggers a strain relaxation process, whereby a mixed-phase state comprising a tetragonal-like ($T'$) phase and a rhombohedral-like ($R'$) phase emerges[32,33]. This mixed-phase BFO system is a fertile ground for intriguing physical properties including large piezoelectric responses[34,35] and field-induced strains[36], electrochromic effects[37], non-zero magnetic moments[38], electrical conductivity[39–41], interesting mechanical properties[42,43], and is a suitable system for reducing the polarization retention loss[30]. Herein, we demonstrate that by intentionally introducing designer defects, macroscale strain coherence is maintained throughout the thickness of the film, which leads to the preservation of the $T'$ BFO phase and thus a complete suppression of the mixed-phase structures throughout the film[44,45]. These defects pervade the entire film uniformly and apply a local compressive strain. This effectively pins the domain walls and impedes domain backswitching as evidenced by a large increase in activation fields for domain wall motion, and thus very long polarization retention is achieved.

## Results

**Structural analysis of the BFO epitaxial thin film.** Thin films of 2% cobalt-doped $T'$ phase BFO thin films with a thickness of 60 nm were grown on a (001) LAO substrates by pulsed laser deposition, with a 3 nm-thick $La_{0.67}Sr_{0.33}MnO_3$ (LSMO) bottom electrode inserted between the substrate and the film (see Methods). The clear atomic steps in a representative topography image in Fig. 1a show a high-quality surface of the thin film and no $R$-like striped domains are found, consistent with previous studies on similar films[44,45]. Figure 1b presents the corresponding IP piezoresponse force microscopy (PFM) phase image. The striped IP domains oriented along <110> -type directions indicate a monoclinic structure ($M_c$)[46–48], in agreement with previous X-ray diffraction (XRD) reciprocal space mapping (RSM) data[44]. In Fig. 1c, the high-angle XRD $\theta-2\theta$ scan exhibits only (00 l) peaks yielding an out-of-plane (OOP) lattice parameter $c = 4.67$ Å. The broad peak at 30.4° has been suggested to be a tetragonal $\beta$-$Bi_2O_3$ phase, which can impose a net compressive strain on BFO and thus stabilize the $T'$ phase (without forming relaxed $R'$ phase striations) at thicknesses well over 60 nm[45,49]. High-angle annular dark field (HAADF) scanning transmission electron microscopy (STEM) was used to study the defect structure in these samples at the atomic scale (Fig. 1d). Clear boundaries for the BFO/LSMO/LAO heterostructure indicate a good quality of epitaxial thin film growth. Defective nanoregions are homogeneously formed throughout the film as shown by yellow-boxed regions in the BFO layer. In Fig. 1e, a HAADF-STEM image of a single defective region is shown and the local change in atomic structure is clearly visible. A clear homogeneous distribution of these defective nanoregions is shown in an annular bright field (ABF)-STEM image in Supplementary Fig. 1. Previous work has demonstrated that nanodomains and domain walls can be pinned by such defects[50,51].

**An objective method for domain diameter analysis.** To explore the ferroelectric retention capabilities of these samples, SPM with conductive tips was used to electrically write nanoscale domains. Domain sizes and diameters are dependent on both tip voltages and pulse durations, which can be used to study the domain wall motion dynamics. Domains were fabricated by applying a voltage across the film between a conductive AFM tip, which serves as a top electrode, and the bottom LSMO electrode. Figure 2a presents the domain diameter dependence on pulse durations at a fixed tip voltage of −9 V with a pulse time ranging from 2.5 ms to 300 ms. The produced arrays of domains with a regular spacing of 300 nm are homogenous and well-defined showing no anomalously nucleated domains. To objectively ascertain the precise value of domain diameters, we use a modified two-dimensional (2D)-Gaussian approach (see Supplementary Note 2 and Supplementary Figs. 2–5 for details) to get azimuthally averaged domain diameters, as schematically shown in Fig. 2b. The upper left image is the raw data of the OOP PFM amplitude image that shows a clear circular domain wall with a low piezoresponse. Two line profiles across the centre of the domain are sketched (white dashed lines) and thus the line segments going through the centre yield the diameter of the domain. Three minima are fitted by the red, purple, and blue curves separately, then a cumulative fit (coloured green) is obtained. The red and blue curves fit the positions for the left and right endpoints of the diameter, whereas the purple fitting curve serves as an offset for normalizing the

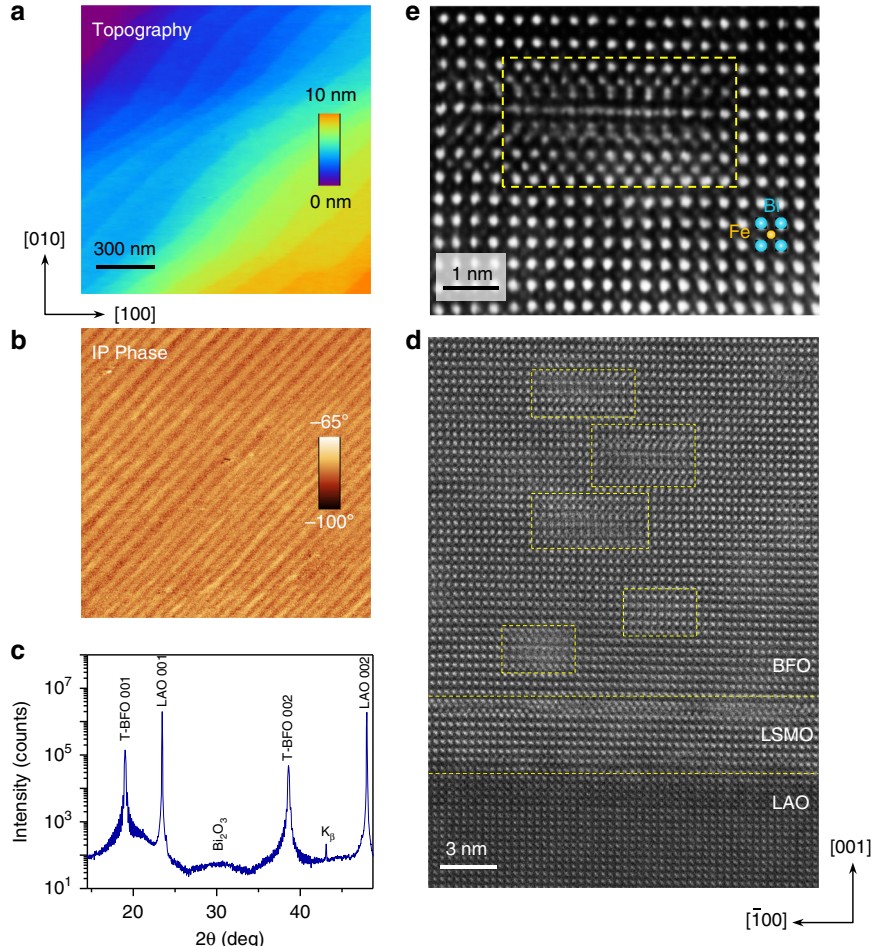

**Fig. 1 Structural analysis of a Co-doped BiFeO₃ epitaxial thin film. a** Surface topography and **b** in-plane piezoresponse force microscopy (PFM) phase images. The striped in-plane domains with a [110] orientation suggest a monoclinic distortion within the tetragonal matrix. **c** High-resolution X-ray diffraction $\theta-2\theta$ scan. **d** High-angle annular dark field (HAADF) scanning transmission electron microscopy (STEM) image of the sample, which shows the distribution of the defects in the film. **e** A HAADF-STEM image of a single defect with a higher magnification (from a different area as compared with **d**) shows the local structure of the defective region (denoted by the yellow dashed box).

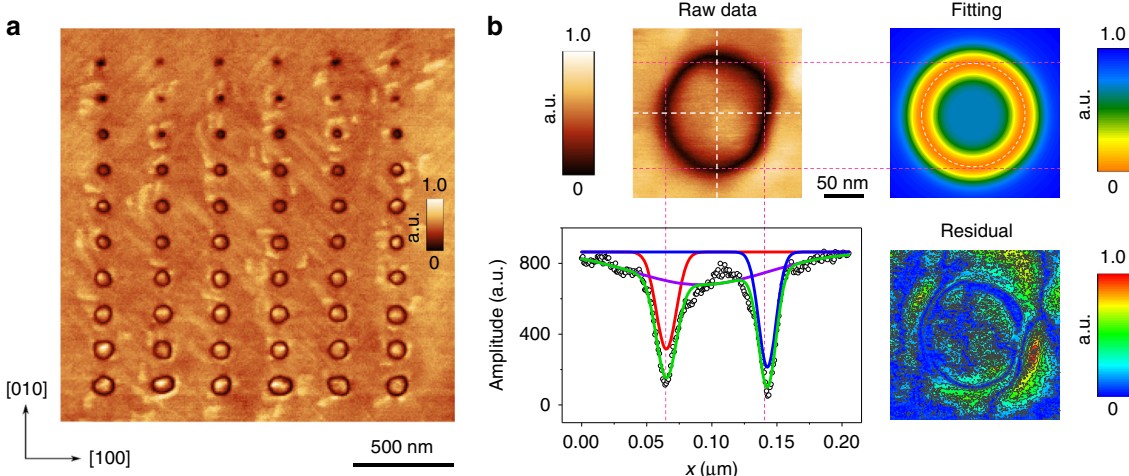

**Fig. 2 PFM out-of-plane amplitude image showing voltage-dependent domain sizes and diameter data analysis method modified 2D-Gaussian fitting method. a** The out-of-plane PFM amplitude image of domains fabricated at −9 V by different pulse time durations (2.5 ms, 5 ms, 10 ms, 20 ms, 40 ms, 50 ms, 100 ms, 200 ms, 300 ms, and 400 ms from the bottom line to the top line, respectively). **b** Illustration for the modified 2D-Gaussian fitting method for calculating domain diameters. (i) Raw data of out-of-plane PFM amplitude image, in which line profiles of the diameter are drawn. (ii) In the corresponding amplitude line-profile, three valleys are fitted by the red, purple, and blue curves, respectively, and a cumulative fit valley is thus achieved (coloured green). (iii) Fitting map of the raw data and (iv) the residual.

piezoresponse inside and outside the domain, which is mostly needed for smaller domains with less clear domain walls. By effectively combining a multitude of line profiles going through the circular domain centre, a 2D fitting contour mapping for the raw data is then obtained (upper right image in Fig. 2b), which further gives an average value for the domain diameter. The residual, the difference between the raw data and the fitting data, which gauges the reliability of the fitting result, shows an almost zero response around the domain wall suggesting a good fitting (lower right image in Fig. 2b). It should be noted that due to the tip resolution, the apparent domain wall width imaged by a sharp tip is around 18 nm, which is far from its real value (on the unit-cell level)[52,53]. Consequently, domain diameters smaller than ~30 nm are not reliably accurate in our measurements.

**Domain wall motion dynamics and activation field.** Domain diameters as a function of tip voltage and pulse duration are statistically analysed in Fig. 3. A representative image of domain diameters in relation to the pulse time duration is shown in

Fig. 2a and Supplementary Fig. 6a presents the domain diameters dependence of tip voltages at a fixed pulse duration of 300 ms. To obtain more statistically reliable results, 22 data sets from experiments in different regions of the film were obtained, in which 12 data sets are for fixed voltages (−8 V, −9 V, and −10 V) and 10 for fixed pulse durations (100 ms, 200 ms, and 300 ms) with details and raw data shown in Supplementary Note 3 and Supplementary Fig. 6. The domain diameters increase logarithmically with increasing pulse durations (Fig. 3a). Although the diameters show a linear dependence on the tip voltages, different fixed pulse durations result in parallel shifts of the fitted lines (Fig. 3b). The domain diameters as a function of pulse duration and voltage reveal a typical size-dependence behaviour[25,54–57]. More quantitative analysis was performed, by calculating the domain wall velocity as a function of the inverse applied electric field for the (Fig. 3c). The data point for each tip voltage are from Fig. 3a. The domain wall velocity can be extracted by dividing the radius increment during two subsequent writing pulse times by the corresponding time interval, as shown in Eq. (1) where $r_{t1}$ and $r_{t2}$ are the domain radius at time $t_1$ and $t_2$, respectively. The local

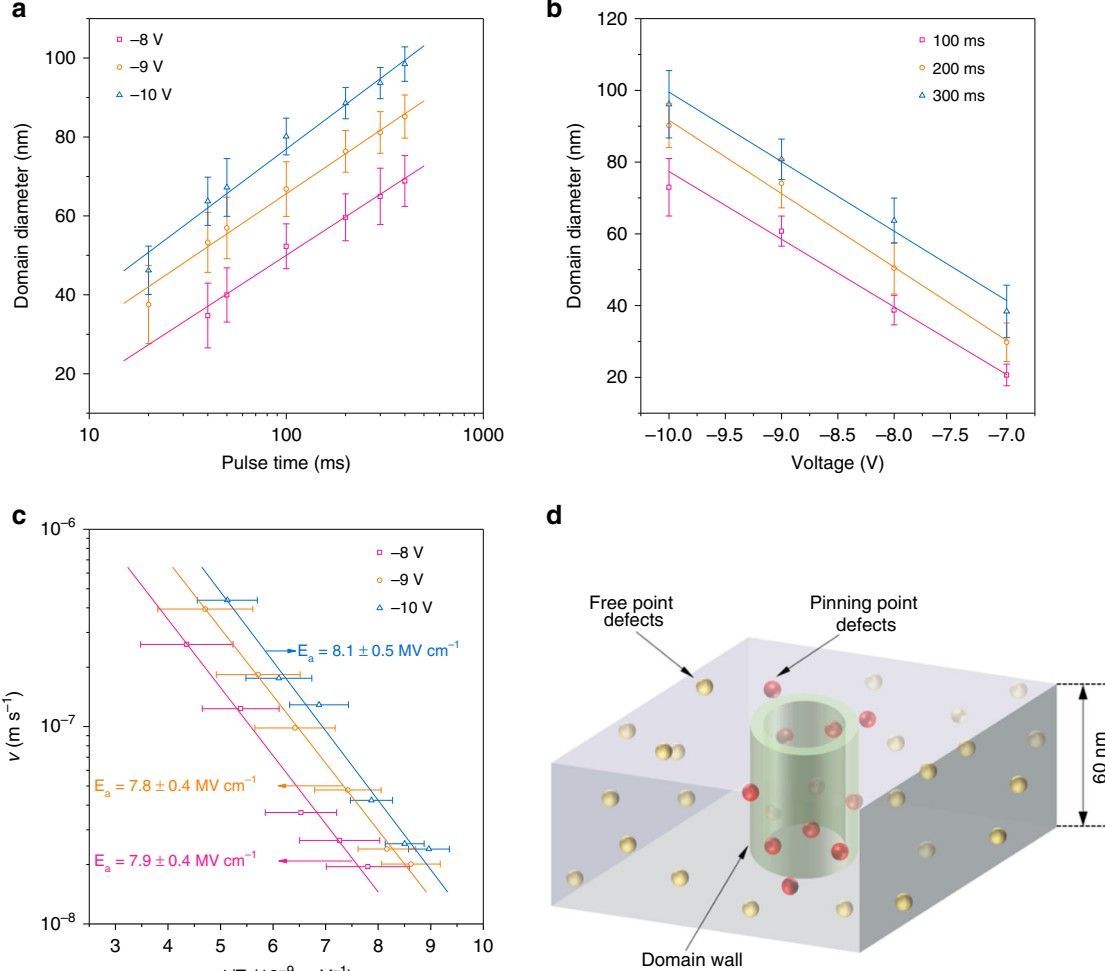

**Fig. 3 Domain diameter as a function of pulse durations and tip voltage, domain wall velocity under various tip voltages, and activation fields.** **a** Domain diameter as a function of different pulse times and **b** different voltages. The data in **a** are fitted logarithmically to the voltage pulse times with error bars. The data in **b** are fitted linearly to tip voltages. The error bars in **a, b** are SDs of domain diameters. **c** Domain wall velocity as a function of the inverse applied field generated by different pulse voltages. The data fit well to Eq. (4) with SDs of $1/E$ as error bars. The activation fields obtained from the fittings are also shown in **c**. **d** A 3D schematic to show a cylindrical domain wall being pinned by point defects that pervade the entire film thickness. Defective nanoregions (red, yellow spheres) act as pinning point defects that can exert local strain to pin domain wall motion. All error bars presented in this figure represent the SDs of the average values of 3~4 data sets at each fixed voltage and pulse duration.

electric field generated by the tip can be determined using Eq. (2) where $V_{tip}$ is the tip voltage, $R_{tip}$ is the radius of the tip, $d_{film}$ is the thickness of the film and $r_{domain}$ is the domain radius[57]. $r_{domain}$ can be expressed as shown in Eq. (3). The domain wall velocity $v$ as a function of the applied electric field follows Eq. (4), where $v_0$ is a parameter with the dimensionality of velocity and $E_a$ is the energy barrier[57].

$$v = \frac{r_{t2} - r_{t1}}{t_2 - t_1} \qquad (1)$$

$$E = \frac{V_{tip} \times R_{tip}}{d_{film} \times r_{domain}} \qquad (2)$$

$$r_{domain} = \frac{r_{t1} + r_{t2}}{2} \qquad (3)$$

$$v = v_0 e^{\left(-\frac{E_a}{E}\right)} \qquad (4)$$

The fitted activation field $E_a$ values for tip voltages of $-8$, $-9$, and $-10\,V$ are $7.9 \pm 0.4$, $7.8 \pm 0.4$, and $8.1 \pm 0.5\,MV\,cm^{-1}$, respectively. Significantly, these activation field $E_a$ values are three to six times larger than the reported $E_a$ values in conventional BFO systems ($1.3$, $1.03$, and $\sim 1.85\,MV\,cm^{-1}$)[58-60]. This significant increase can be attributed to the local defects (as shown by STEM in Fig. 1d) that effectively pin the domain walls and thus higher energy is needed to overcome the energy barrier and unpin the domain walls. As we show in the following, this particularly high value of activation field is consistent with the concept that these defects enable very long polarization retention. Furthermore, from the STEM image (Fig. 1d and Supplementary Fig. 1) and the previously reported TEM results[45], we can estimate the density of the defective nanoregions. The average size of the defective

nanoregion is $\sim 5\,nm$ in width and $\sim 2\,nm$ in height with a density of 48 nanoregions $1659\,nm^{-2}$. Based on such an analysis, even for a small cylindrical domain with a diameter of 20 nm, the domain wall traverses about 40–50 of these defects, which suggests that the density of these defective nanoregions is high enough to provide effective pinning of the proximal domain wall. For larger domains, therefore, based on the increase in domain wall length, considerably more pinning centres are involved. More details are discussed in Supplementary Note 1 and Supplementary Fig. 1. A schematic is shown in Fig. 3d, in which defective nanoregions are referred to as point defects represented by red and yellow spheres. In the three-dimensional space, many point defects (in red) are pinning the cylindrical domain wall, while other free point defects (in yellow) are homogeneously distributed throughout the entire film thickness.

**Ultra-long polarization retention.** The SPM-based polarization retention experiments were conducted by applying dc voltage pulses to a conductive tip to switch ferroelectric domains and examine the domain sizes or other polarization-related properties (e.g., piezoresponse, surface potential, and polarization values) over timescale from minutes to days. Domains of various sizes were written at a fixed tip voltage of $-9\,V$ at different pulse durations ranging from 5 ms to 200 ms. High-resolution OOP PFM amplitude images were then recorded sequentially from $t = 0\,h$ to $t = 8900\,h$ (i.e., spanning a time window of more than one year), which is shown in Fig. 4a. The amplitude of ac excitation in the PFM images was 2.0 V, which is lower than the threshold to switch the polarization during the readout. Diameters for domains of all sizes, albeit large or small initially, remain almost the same (Fig. 4b). In Supplementary Fig. 7, another data set with domains fabricated in a different area shows the same almost no-

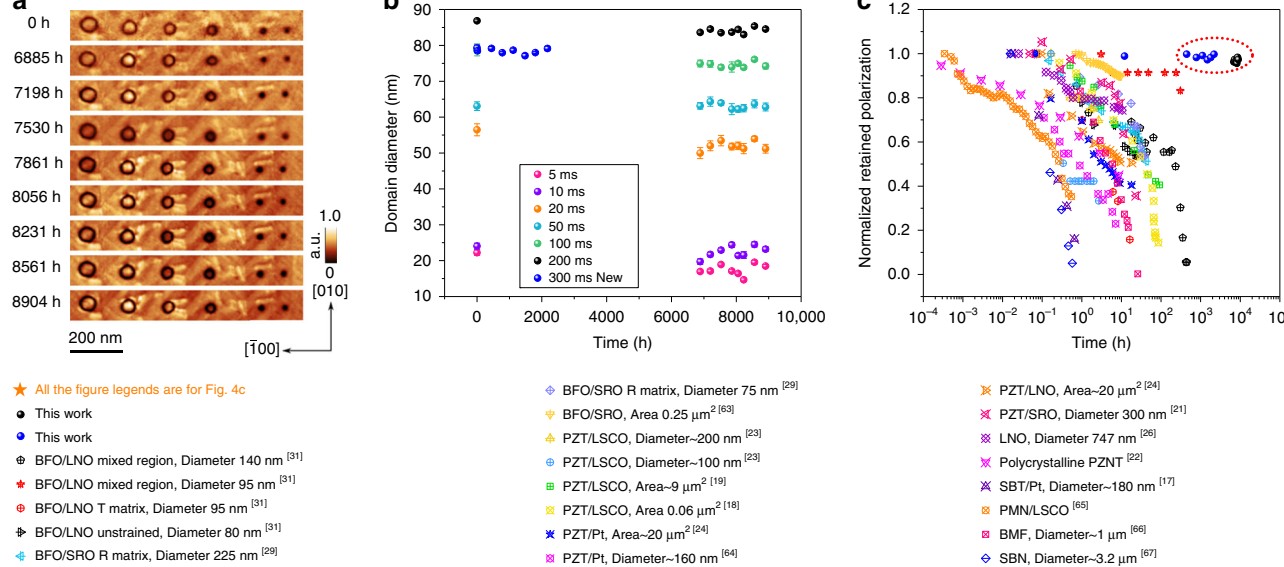

Fig. 4 Polarization retention over a duration of >1 year showing no degradation over time in comparison to other ferroelectric systems. a Out-of-plane PFM amplitude images of domains of different diameters recorded at the times indicated. The domains were fabricated by a tip voltage of $-9\,V$ with different pulse duration times (from right to left: 5 ms, 10 ms, 20 ms, 50 ms, 100 ms, and 200 ms). The needle-like rhombohedral variants are formed around the tetragonal domains concomitantly during the switching, as observed previously[61]. b Domains of various diameters and their change over time spanning 8904 h showing excellent polarization retention. The data are extracted from a and fitted by the modified 2D-Gaussian method. In another separately poled region, the domains poled with 300 ms at $-9\,V$ with a different tip display a no-decay trend of the polarization during the first 2000 h. The error bar shown here represents the standard error for each data point fitted by the modified 2D-Gaussian method. c A comparison of normalized retained polarization versus relaxation time between $BiFeO_3$ thin film in this study and other ferroelectric materials in previous reports, showing superior retention performance. The solid blue data points show the retention behaviour in the preliminary 2100 h and the solid black data points show the retention behaviour over 8900 h. Please note that all the legends shown below the figures are for Fig. 4c.

decay trend (e.g., 79.4 nm at $t = 0$ h and 79.1 nm at $t = 8900$ h). In addition, concomitant with the switched domains, some bright needle-like structures that exhibit higher piezoresponse than the matrix are formed around. These are rhombohedral variants whose formation is due to collective interactions of tip-induced electric field and elastic strain[61]. The time-dependent evolution of domain sizes spanning over 8900 h is shown in Fig. 4b. All the diameters with different sizes are extracted from Fig. 4a and data points are individually fitted by the modified 2D-Gaussian method. A separate data set that spans the preliminary 2100 h is also shown in Fig. 4b. It is to be observed that since this data set is taken from a separately poled region with different tip conditions, the domain sizes are different from the original poled region. All the data points have error bars that are negligible in comparison to the diameter of the data points. As can be seen in the figure, diameters of all domain sizes are nearly constant over time within error bars. The small variations from image to image are likely caused by slightly different imaging conditions for each measurement, considering the large time span over which measurements were conducted. A linear fit is used to fit all the data sets in Fig. 4(b) and the slopes for all the fitted lines are almost zero (at the 0.05 level, the slope is not significantly different from zero). Therefore, we can assume that domains of all studied sizes do not experience any significant polarization degradation. For instance, for the domain written by $-9$ V for 200 ms, the diameters are 86 nm and 85 nm at $t = 0$ h and $t = 8900$ h, respectively; for the domain written by $-9$ V for 50 ms, the diameter maintains 63 nm from $t = 0$ h to $t = 8900$ h. The differences seen here are minuscule, whereas for the smallest domain written by $-9$ V for 5 ms the diameter changes from 22 nm at $t = 0$ h to 19 nm at $t = 8900$ h. It is noteworthy that due to the tip resolution, the diameter values smaller than 30 nm are not reliably precise; therefore, a larger deviation is expected for the data set of 22 nm domains (details discussed in the Supplementary Note 2). Temperature stability for the polarization retention was also studied (see Supplementary Note 4 and Supplementary Fig. 8). It was found that up to 175 °C, the polarization is still very robust, with the average diameter of the domains only decreasing by around 10%. Such stable polarization retention behaviour over a relatively wide temperature range is of significance for electronic information storage applications.

Figure 4c shows a comparison of normalized polarization retention time between our BFO system and other ferroelectric systems[16–18,20–23,25,28,30,62–66]. The decay in most systems follows either an exponential function[21,23], a log time[16,21,23] or a three-stage trend[25,30] (decreasing quickly-reaching a stable platform-decreasing quickly again). Remarkably, our BFO system stands out by at least one to two orders of magnitude. It is also noteworthy that the polarization retention in our BFO system does not show a dependence on domain size, i.e., the polarization states are extremely stable for all domain sizes, even for domains with diameters around 20 nm. Such an observation is at odds with the typical observation that larger domains yield longer retention times, as a larger switching area means a slower relaxation process[16,17,20,22,25,30,67]. However, at the same time, larger domains are less attractive for high-density memory applications. In our work, based on the smallest domain diameter, the estimated storage density is 158 Gbit inch$^{-2}$ and a density of 1300 Gbit inch$^{-2}$ can be possibly achieved in the future with sharper tips. Notably, the polarization retention time for the domains written on the $T'$ matrix in our defect-engineered BFO sample, which is tested until $t = 8900$ h, is so far 177 times longer than that for the domains written on the $T'$ matrix in a normal strained BFO sample with a retention time around 50 h (written by $-12$ V with a pulse duration of 100 ms), as reported and shown in Fig. 4c[30].

## Discussion

In our BFO system, a defect engineering method is adopted to solve the retention problem. As mentioned above, in this T-like BFO film, nanoscale defective regions are homogeneously distributed, maintaining macroscale strain coherence throughout the entire film thickness. After the creation of a stable nucleus below the tip, polarization switching proceeds with domain wall motion and domain walls can be regarded as interfaces moving through a disordered medium[68,69]. The nanoscale defect region in our case is the dominant mechanism for pinning the domain walls[45]. Also, in Supplementary Figs. 9 and 10, it is shown that the $R'/T'$ phase boundary (referred to as $R/T$ in ref. 30) is not the dominant factor for pinning the domain walls and thus enhancing the retention in our BFO system. After 8921 h, neither domain sizes for domains grown on $T'$ matrix and $R'/T'$ mixed-phase boundaries changed significantly (98% retained for domains on $T'$ matrix and 94% retained for domains of $R'/T'$ mixed phase). The diameters for domains grown on $R'/T'$ mixed-phase boundaries even decrease slightly. The reason for this might be due to the fact that the electrically written $R'$ phases are not energetically favourable (as they are not the ground state of the system) and therefore tend to switch back to pure $T'$ phase over time (for further details, see Supplementary Note 5). Consequently, the local strain will be partially released when electrically written $R'$ domains undergo backswitching.

We have thus demonstrated extremely robust nanoscale ferroelectric domains that remain almost unchanged over 1 year in defect-engineered $T'$-BFO films. Extrapolation of the domain sizes yields an expected lifetime of well over 10 years. The origin for such long polarization retention is proposed to be the result of designer-defect nanoregions which are present throughout the volume of the film. These nanoregions, formed during the film fabrication, preserve strain coherence throughout the film. Locally, the strain provided by these point defects can pin the domain walls[70] from a polarization reversal driven by an internal field due to asymmetric electrostatic boundary conditions. We find that domain wall movement is inhibited, as shown by significantly increased activation fields for domain wall movement. In addition, the polarization retention behaviour is independent of the initial domain size, a feature that is highly desirable towards the goal of increasing storage density while maintaining a good retention. The $T'$- BFO system therefore is a very promising material for high-density memory devices with extremely long retention times (for instance in archival storage). In addition, our work offers generic guidelines for improving polarization retention in ferroelectrics using defect engineering, offering strategies for overcoming limits for practical nanoelectronics applications of ferroelectric thin films.

## Methods

**Thin film growth.** Epitaxial Co-doped BFO/LSMO/LAO thin film heterostructure were deposited on (001) oriented LAO substrate by pulsed laser deposition (KrF $\lambda = 248$ nm excimer laser, fluence ~4 J cm$^{-2}$, laser repetition rate 3 Hz) using a ceramic Bi$_{1.1}$Fe$_{0.98}$Co$_{0.02}$O$_3$ target, and the target (substrate-target distance ~ 10.5 cm) was held at 650 °C at a deposition rate of ~0.015 Å per pulse. The samples were cooled down at 20 °C min$^{-1}$ in 450 Torr of oxygen. The BFO film is around 60 nm. A 3 nm-thick LSMO was inserted between the film and the substrate as the bottom electrode with growth conditions of 800 °C and 100 mTorr of oxygen.

**X-ray diffraction.** High-resolution $\theta-2\theta$ XRD and RSM studies were carried out using a PANalytical X'Pert Pro diffractometer with CuK$_{\alpha-1}$ radiation.

**Scanning probe microscopy.** Surface morphology and ferroelectric domains were observed by a commercial AFM system (AIST-NT Smart SPM 1000) under ambient conditions. Nanodomains of various sizes were created by applying pulse voltages on the film by conductive platinum coated tips (Mikromasch HQ:NSC35/Pt) with sample grounded. The evolution of domain structures was recorded by

PFM with an ac excitation amplitude of 2.0 V. Samples were stored in air at room temperature and humidity below 40% for the duration of the study.

**Transmission electron microscopy**. HAADF-STEM and ABF-STEM were carried out on a dual Cs-corrected (probe and image) FEI Titan[3] 80–300 FEGTEM operated at 305 kV with a convergence angle of 15 mrad. The STEM specimen was prepared with focused ion beam method on a FEI Quanta 3D FEGSEM.

## Data availability

All data used are available within this manuscript and Supplementary Information. Further information can be acquired from the corresponding authors upon reasonable request.

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

## Acknowledgements

We acknowledge support by the Australian Research Council through Discovery Grants and the ARC Centre of Excellence in Future Low Energy Electronics Technologies (FLEET). D.Z. acknowledges an Australian Government Research Training Program Scholarship. The research at Monash Centre for Electron Microscopy (MCEM) used equipment funded by Australian Research Council Grant Number LE0454166 (FEI Titan[3] 80–300 FEGTEM). We thank MCEM for the provision of equipment and technical support. We acknowledge Thomas Young and Vicki Zhong for assistance with sample preparation.

## Author contributions

J.S., P.S., and D.Z. conceived and designed the experiments. D.S. and V.G. carried out sample preparation and XRD characterization. D.Z. conducted the SPM measurements. X.C. and M.W. performed the TEM experiment. D.Z., F.J., D.S., P.S., N.V., and J.S. analysed the data. D.S., P.S., and J.S. supervised the project. D.Z. and D.S. led the manuscript preparation with contributions from all authors.

## Competing interests

The authors declare no competing interests.
