## [Peer Review File · Nature Communications]

REVIEWERS' COMMENTS:

Reviewer #1 (Remarks to the Author):

In their paper Superior polarization retention through engineered domain wall pinning Zhang et al. study retention in T phase of Co doped BFO on LSMO on LAO. The topic is highly relevant, as BFO, a promising multiferroic, often is prone to ferroelectric back-switching and remedies are highly sought after. Albeit the study relies on standard techniques, and a model to extract domain wall velocity from PFM data from the early 2000s, the study is an example on how relatively simple techniques used in a very good fashion, in combination with new analysis tools as the presented 2D-Gaussian method can give novel insight into important problems. I believe the study merits publication in Nature Communications after two issues have been addressed:

1. The authors perform the study on 60nm 2% Co doped BFO on LSMO on LAO. They claim that the Co-doping allows to synthesize T' only thin films by this doping, as often a combined R/T phase thin film are obtained without. However, one of the main arguments of the paper is that the defective nano-regions observed are responsible for 'hard' pinning and hence very little retention. In the supplementary information they also show data for a R'/T' sample showing slightly better retention in the T' phase as compared to the R' phase, but more retention as compared to the thick film presented in the main paper. In order to more directly couple the very stable polarization states to the Co-doping and nanoregions it would benefit the paper if the authors could comment on:
 - a. Have they performed any chemical composition of the nanoregions, and structural analysis?
 - b. If thinner samples were studied, below the 30nm limit to grow pure T', is the retention then lower without the Co doping as compared to doped samples as studied here?
2. In Figure 4, for clarity, it would help if it was written that all the listed correspond to fig. 4c. This since similar colors are used also in fig. 4b.

Reviewer #2 (Remarks to the Author):

In this paper the authors present an interesting result where defect pinning in BiFeO₃ thin films lead to greater domain stability. I think their basic finding is solid and interesting, but I find there are substantial improvements they should make in their analysis of the domain growth under field.

The first problem is that they describe the parameter E_a several times as an activation energy. It is not an energy, it is plainly a field, as even the units they use for it make clear. This is easy to fix as for all intents and purposes except describing it they have treated it as a field, but I'm a bit surprised they made this mistake in the first place.

In Ref 57, cited as the origin of the model being used, there was in fact a dynamical exponent μ in the equivalent of equation 4. That paper discusses the significance of that parameter being close to 1 at great length, here it is simply ignored. Did the authors consider different values of μ , or have any comments on this?

The supplementary material goes in to great lengths about the methods used to fit the domains and describe how the error bars in Fig 3 are arrived at. This is nice, but it's quite disappointing that it then appears that these error bars have been completely neglected in coming up with the activation field values. The uncertainty in these values should be considerable (my rough estimate is about +/- 2MV/cm) and should be calculated and included with the stated values.

If I put these concerns aside there is no doubt about the basic results: the domains in these films are more stable and they are pinned to the point that they move much more slowly under an electric field. Therefore I would like the authors to fix the issues described above, after which I feel this paper will be appropriate for publication in Nature Communications.

Reviewer #3 (Remarks to the Author):

The manuscript by D. Zhang et al. presents a study of ultra-long polarization retention in intentionally engineered domains with conductive SPM tips in Co doped BFO thin films. The key point of the work is the engineering of defect regions that pin domain walls and thus elevate their activation energy preventing the degradation in switched nanoscale domains for periods longer than one year. The obtained results approach the application of ferroelectric materials in data storage applications, however, some clarification is needed for publishing in Nature Communications journal.

1. Point defects are the result of designer-defect-engineering method of BFO thin film growth. It would be logical to highlight the details of film preparation leading to such defects formation or add the corresponding reference to the Methods section.

2. The strain in ferroelectric films strongly influences phase transition temperatures. There is no information in the manuscript about the evolution of the domain structure following the temperature change that is important for applications. In what temperature range are domain walls able to keep such retention according to the phase diagram?

3. In Fig.1 d the enlarged part has quite a low resolution. It is highly desirable to show in more details the structure of the defect (e.g. similar sketch of the lattice in disordered region with comments).

4. In the manuscript only 60nm thin films are discussed. It would be useful to comment the behavior of the film phase, defect distribution and domain retention for different thicknesses.

Reply to reviewer comments

We would like to express our gratitude to all the reviewers for their detailed and insightful comments, which prompted us to make a number of changes to the manuscript as described below, resulting in a substantially improved version. Herein, we have addressed all the comments (in black) with a point-by-point response (in blue). Added text in the manuscript is shown in red colour.

Reviewer #1 (Remarks to the Author):

In their paper Superior polarization retention through engineered domain wall pinning Zhang et al. study retention in T phase of Co doped BFO on LSMO an LAO. The topic is highly relevant, as BFO, a promising multiferroic, often is prone to ferroelectric back-switching and remedies are highly sought after. Albeit the study relies on standard techniques, and a model to extract domain wall velocity from PFM data from the early 2000, the study is an example on how relatively simple techniques used in a very good fashion, in combination with new analysis tools as the presented 2D-Gaussian method can give novel insight into important problems. I believe the study merits publication in Nature Communications after two issues have been addressed:

Our Response:

We thank the reviewer for finding our work relevant and interesting.

1. The authors perform the study on 60nm 2% Co doped BFO on LSMO on LAO. They claim that the Co-doping allows to synthesize T' only thin films by this doping, as often a combined R/T phase thin film are obtained without. However, one of the main arguments of the paper is that the defective nano-regions observed are responsible for 'hard' pinning and hence very little retention. In the supplementary information they also show data for a R'/T' sample showing slightly better retention in the T' phase as compared to the R' phase, but more retention as compared to the thick film presented in the main paper. In order to more directly couple the very stable polarization states to the Co-doping and nanoregions it would benefit the paper if the authors could comment on:

a. Have they performed any chemical composition of the nanoregions, and structural analysis?

Our Response:

We have performed electron energy loss spectra (EELS) at the non-defect regions and defective regions, as shown in Fig R1. We could not detect any chemical difference between them within

detection limits of the equipment and the specimen beam damage. Given that the size of the defect (approximately 3 nm) is only 15% of the average thickness of the transmission electron microscopy specimen (around 20 nm), the EELS signal is dominated by the signal from the non-defective material. This is the likely reason why no chemical composition changes could be detected, it is also possible that these regions are largely structurally different, not chemically different. It should be noted that only oxygen peaks are shown in the EELS data because no changes or energy losses for Fe and Bi can be detected within the sensitivity limit of the experiment.

Fig R1 Local chemical analysis for defective nanoregions. a High angle annular dark field scanning transmission electron microscopy (HAADF-STEM) image indicates the locations where electron energy loss spectra (EELS) were performed. **b** EELS oxygen K-edges acquired from the defected site as well as bulk region (indicated by red rectangular)

b. If thinner samples were studied, below the 30nm limit to grow pure T' , is the retention then lower without the Co doping as compared to doped samples as studied here?

Our Response:

According to the reviewer's suggestion, two more 20 nm T phase BFO samples without Co doping were fabricated and their polarization retention behaviour was studied in comparison to the thicker 60 nm Co doped BFO sample, one standard T phase BFO sample and one defect engineered T phase BFO sample. The 20 nm defect engineered sample was grown under the same conditions as the 60 nm defect engineered BFO sample to ensure the existence of the same kind of defects with similar defect density. Both two 20 nm T phase BFO samples were predominant T phase.

Polarization retention experiments ranging from 0 day to 8 days were performed on both samples, as shown in Fig. R2. The results show that the polarization of the 20 nm BFO sample with the engineered defect is very robust as the diameters for domains switched by different pulse durations are almost identical spanning a time scale of 8 days. In contrast, the 20 nm standard *T* phase BFO sample shows clear polarization decay within the same time scale as the diameters for all sizes of domains are shrinking, and some of the domains have shrunk significantly. Following our earlier argumentation and model, domains in 20 nm standard *T* phase BFO vanish eventually and domains in 20 nm *T* phase BFO with engineered defects maintain their size and behave similarly as the 60 nm BFO sample with defects.

Therefore, our conclusion is that for the thinner sample, engineered defects can effectively pin the domain walls and the retention behaviour is comparable to that of the thicker sample. The Co doping is not necessary for the long retention behaviour.

Fig R2 Polarization retention behaviour for two thinner *T* phase BFO samples with a thickness of 20 nm. The out-of-plane PFM images for 0 day and 8 days are recorded for 20 nm standard *T* phase BFO sample and 20 nm defect engineered *T* phase BFO sample, respectively. The scale bar is 200 nm and it applies to all figures in Fig R2.

2. In Figure 4, for clarity, it would help if it was written that all the listed correspond to fig. 4c. This since similar colors are used also in fig. 4b.

Our Response:

We thank the reviewer for this helpful suggestion to improve our manuscript. We have modified Fig. 4 by adding one more legend at the beginning of all the legends to claim that “all the figure legends are for Fig. 4c” (as shown in Fig. R3). Besides, in the caption for Fig. 4, we have added one sentence “Please note that all the legends shown below the figures are for Fig. 4c”.

Fig R3 Polarization retention over a duration of > 1 year showing no degradation over time in comparison to other ferroelectric systems. a Out-of-plane PFM amplitude images of domains of different diameters recorded at the times indicated. **b** Polarization retention behaviour comparison between different systems.

Reviewer #2 (Remarks to the Author):

In this paper the authors present an interesting result where defect pinning in BiFeO3 thin films lead to greater domain stability. I think their basic finding is solid and interesting, but I find there are substantial improvements they should make in their analysis of the domain growth under field.

Our Response:

We thank the reviewer for the positive comments and suggestions.

The first problem is that they describe the parameter E_a several times as an activation energy. It is not an energy, it is plainly a field, as even the units they use for it make clear. This is easy to fix as for all intents and purposes except describing it they have treated it as a field, but I'm a bit surprised the made this mistake in the first place.

Our Response:

The reviewer is right, and we have made changes accordingly. E_a is the critical field above which the domain walls are depinned by the defects. Thus, we have replaced all the text parts with “activation energy” by “activation field” in the main text (marked in red in the main text). We thank the reviewer for pointing this out.

In Ref 57, cited as the origin of the model being used, there was in fact a dynamical exponent μ in the equivalent of equation 4. That paper discusses the significance of that parameter being close to 1 at great length, here it is simply ignored. Did the authors consider different values of μ , or have any comments on this?

Our Response:

We indeed considered other values for this dynamical exponent μ and found that only the value of $\mu=1$ can lead to a successful fitting.

This dynamical exponent μ reflects the dimensionality of the wall and the nature of the pinning potential. If the defects can only locally modify the symmetric ferroelectric double-well potential depth, it is a “random bond” scenario where $\mu=0.25$ for one-dimensional walls and $\mu\sim 0.5-0.6$ for two-dimensional walls. By contrast, if the defects induce a local field that can break the symmetry of the ferroelectric double-well potential, it is a “random field” scenario where $\mu=1$ regardless of the domain wall dimensionality.

Here we take the velocity versus $1/E$ written by a tip voltage of -8 V for instance, and we use the equation $V = V_0 e^{-\left(\frac{E_0}{E}\right)^\mu}$ to fit this data set. When we fix $\mu=1$, it is a perfect fit which gives an activation field value of $E_a=7.9\pm 0.4$ MV/cm with an R-Square (COD) value of 0.99623 indicating a very good and reliable result. By contrast, we have tried other μ values including 0.25, 0.26, 0.5, 0.6, 0.9 and 1.1, but none of the fitting parameters worked and fits did not converge in these cases.

The linear fittings in Fig. 3c indicate a dynamical exponent $\mu\sim 1$. This μ value suggests that the defects induce a long-range local field to pin the domain walls, which corresponds to the “random field” scenario. This also corroborates the existence of defective nanoregions that are homogeneously distributed in the film to pin the domain walls.

The supplementary material goes into great lengths about the methods used to fit the domains and describe how the error bars in Fig 3 are arrived at. This is nice, but it's quite disappointing that it then appears that these error bars have been completely neglected in coming up with the activation field values. The uncertainty in these values should be considerable (my rough estimate is about +/- 2MV/cm) and should be calculated and included with the stated values.

Our Response:

We agree that error bars should have been included in Fig. 3. We have added them now, the error bar values are around ± 0.5 MV/cm as shown in Fig. R4.

We would like to note that the error bars of the data points shown in Fig. 3 mainly come from the standard deviation of the average values of at least 4 data sets at each fixed voltage and pulse duration. The average standard deviation value is around 6 nm. However, the error bars shown in Fig. 4b originate from the ‘modified 2D-Gaussian’ fitting approach, and the average value is around 0.8 nm, i.e. much smaller than that of the standard deviation obtained from several datasets. Therefore, the error bars from the standard deviation can serve as an upper limit and rather conservative estimate of error bars. In addition, we have used the time we had during the revision process to update Fig. 3a and Fig. 3c by incorporating more data sets and more reliable statistics, which presents the same trend as before.

Fig. R4 Domain diameter as a function of pulse durations and tip voltage, domain wall velocity under various tip voltages and activation fields.

If I put these concerns aside there is no doubt about the basic results: the domains in these films are more stable and they are pinned to the point that they move much more slowly under an electric field. Therefore I would like the authors to fix the issues described above, after which I feel this paper will be appropriate for publication in Nature Communications.

Our Response:

We thank the reviewer for the valuable suggestions and the recognition of our work.

Reviewer #3 (Remarks to the Author):

The manuscript by D. Zhang et al. presents a study of ultra-long polarization retention in intentionally engineered domains with conductive SPM tips in Co doped BFO thin films. The key point of the work is the engineering of defect regions that pin domain walls and thus elevate their activation energy preventing the degradation in switched nanoscale domains for periods longer than one year. The obtained results approach the application of ferroelectric materials in data storage applications, however, some clarification is needed for publishing in Nature Communications journal.

Our Response:

We thank the reviewer for the positive comments, and we appreciate the valuable suggestions.

1. Point defects are the result of designer-defect-engineering method of BFO thin film growth. It would be logical to highlight the details of film preparation leading to such defects formation or add the corresponding reference to the Methods section.

Our Response:

We agree with the reviewer's suggestion, so we have added the details of the film fabrication and corresponding reference in the Methods section. The modified part in Methods in the main text is marked in red, as shown below:

“Thin Film Growth: Epitaxial Co doped BiFeO₃/La_{0.67}Sr_{0.33}MnO₃/LaAlO₃ thin film heterostructure were deposited on (001) oriented LaAlO₃ (LAO) substrate by pulsed laser deposition (KrF $\lambda=248$ nm excimer laser, fluence ~ 4 J/cm², laser repetition rate 3 Hz) using a ceramic Bi_{1.1}Fe_{0.98}Co_{0.02}O₃ target, and the target (substrate-target distance ~ 10.5 cm) was held at 650 °C at a deposition rate of ~ 0.015 Å per pulse. The samples were cooled down at 20 °C/min in 450 Torr of oxygen. The BFO film is around 60 nm. A 3-nm thick La_{0.67}Sr_{0.33}MnO₃ (LSMO) was inserted between the film and the substrate as the bottom electrode with a growth condition of 800 °C and 100 mTorr of oxygen. For

more details, especially regarding the origin of the defect formation in these particular films, please refer to reference 45.”

2. The strain in ferroelectric films strongly influences phase transition temperatures. There is no information in the manuscript about the evolution of the domain structure following the temperature change that is important for applications. In what temperature range are domain walls able to keep such retention according to the phase diagram?

Our Response:

We agree with the reviewer that the temperature stability of the written domains is also of importance for practical electronic information storage applications. Therefore, we have performed polarization retention experiments at various temperatures as follows. First, nanoscale domains were written by SPM tips at room temperature. Next, the sample was heated to a given target temperature and held at this temperature for 30 min. The sample was then cooled down to room temperature and polarization retention property was checked using the out-of-plane PFM signal. As can be seen from Fig. R5a, the polarization retention is present from room temperature (25 °C) up to 175 °C with very clear domain structures. Temperatures higher than 175 °C may cause chemical changes to the sample so in this experiment we fixed the upper temperature to be 175 °C. It should be noted that the diameters slightly decrease with the increase of the temperature, as shown in Fig. R5b. Here the average diameters of the six domains are the data points and their standard deviations are the error bars. The average decrease of the diameters/polarization at 175 °C is around 10 percent compared to that of the domains at 25 °C. This shows that the polarization retention is quite stable within a wide temperature range.

This figure has been included in the Supplementary Information as Supplementary Figure 7. Correspondingly, we added the text below in the main text and in the Supplementary Information.

In the main text:

“Temperature stability for the polarization retention was also studied (Supplementary Fig. 8). It was found that up to 175 °C, the polarization is still very robust, with the average diameter of the domains only decreasing by around 10%. Such stable polarization retention behaviour over a relatively wide temperature range is of significance for electronic information storage applications.”

In the Supplementary Information:

“The temperature stability of the written domains is also of importance for practical electronic information storage applications. Therefore, we have performed polarization retention experiments

at various temperatures as follows. First, nanoscale domains were written by SPM tips at room temperature. Next, the sample was heated to a given target temperature and held at this temperature for 30 min. The sample was then cooled down to room temperature and polarization retention property was checked using the out-of-plane PFM signal. As can be seen from Fig. S8 (a), the polarization retention can be maintained from room temperature (25 °C) up to 175 °C with very clear domain structures. Temperatures higher than 175 °C may cause chemical changes to the sample so in this experiment we fixed the upper temperature to be 175 °C. It should be noted that the diameters slightly decrease with the increase of the temperature, as shown in Fig. S8 (b). Here the average diameters of the six domains are the data points and their standard deviations are the error bars. The average decrease of the diameters/polarization at 175 °C is around 10 percent compared to that of the domains at 25 °C. This shows that the polarization retention is quite stable over a wide temperature range.”

Fig. R5 Polarization retention at different temperatures. **a** Polarization retention behaviour after being heated at various temperatures. **b** The average diameters as a function of temperature. The scale bar is 200 nm.

3. In Fig.1 d the enlarged part has quite a low resolution. It is highly desirable to show in more details the structure of the defect (e.g. similar sketch of the lattice in disordered region with comments).

Our Response:

We have replaced the enlarged part (high angle annular dark field scanning transmission electron microscopy, HAADF-STEM) by a higher magnification image of a different area to better show the details of the local defective area in Fig. 1 in main text, which is shown in Fig. R6. Again, given the limited size of the defect, the signals in TEM images are dominated by the overlapped non-defect structures and it is non-trivial to generate a realistic model. We expect this would take six months of

microscopy, simulation and DFT calculation to determine a structure with any certainty, which is beyond the scope of the present manuscript.

Fig. R6 Structural analysis of a Co-doped BiFeO₃ epitaxial thin film. **a** Surface topography and **b** In-plane piezoresponse force microscopy (PFM) phase images. The striped in-plane domains with a [110] orientation suggest a monoclinic distortion within the tetragonal matrix. **c** High-resolution X-ray diffraction θ - 2θ scan. **d** High angle annular dark field (HAADF) scanning transmission electron microscopy (STEM) image of the sample, which shows the distribution of the defects in the film. **e** A

HAADF-STEM image of a single defect with a higher magnification (from a different area as compared to **d**) shows the local structure of the defective region (denoted by the yellow dashed box).

The corresponding change in the main text is shown below:

“In Fig. 1e, a HAADF-STEM image of a single defective region is shown, and the local change in atomic structure is clearly visible.”

4. In the manuscript only 60nm thin films are discussed. It would be useful to comment the behavior of the film phase, defect distribution and domain retention for different thicknesses.

Our Response:

We have additionally studied 20nm thick samples, both with and without defects, confirming the trend and arguments made in the paper. For details please kindly see the response to Comment 1b of Reviewer #1.

REVIEWERS' COMMENTS:

Reviewer #2 (Remarks to the Author):

In my opinion the authors have done a thorough job of addressing the concerns raised by myself and the other referees, and the paper is now fully appropriate for publication in Nature Communications.

Reviewer #3 (Remarks to the Author):

Corrections made by authors are clear and sufficient for the publication in Nature Communications journal.

Reviewer #2 (Remarks to the Author):

In my opinion the authors have done a thorough job of addressing the concerns raised by myself and the other referees, and the paper is now fully appropriate for publication in Nature Communications.

Our Response:

We thank the reviewer for the positive comments.

Reviewer #3 (Remarks to the Author):

Corrections made by authors are clear and sufficient for the publication in Nature Communications journal.

Our Response:

We thank the reviewer for the positive comments.